# Effects of In-Classroom Physical Activity Breaks on Children’s Academic Performance, Cognition, Health Behaviours and Health Outcomes: A Systematic Review and Meta-Analysis of Randomised Controlled Trials

**DOI:** 10.3390/ijerph19159479

**Published:** 2022-08-02

**Authors:** D. L. I. H. K. Peiris, Yanping Duan, Corneel Vandelanotte, Wei Liang, Min Yang, Julien Steven Baker

**Affiliations:** 1Department of Sport, Physical Education and Health, Faculty of Social Sciences, Hong Kong Baptist University, Hong Kong, China; hashi@kln.ac.lk (D.L.I.H.K.P.); wliang1020@hkbu.edu.hk (W.L.); 20482450@life.hkbu.edu.hk (M.Y.); jsbaker@hkbu.edu.hk (J.S.B.); 2Centre for Health and Exercise Science Research, Hong Kong Baptist University, Hong Kong, China; 3Physical Activity Research Group, School of Health, Medical and Applied Sciences, Central Queensland University, Rockhampton 4701, Australia

**Keywords:** classroom, physical activity, RCT, theory-based, process evaluation, risk of bias

## Abstract

In-Classroom physical activity breaks (IcPAB) are a promising way to promote children’s health behaviors, while contributing to the development of their academic and cognitive ability and health outcomes. Yet the effect of the activity breaks, which are exclusive to classroom settings, are still mixed and unclear. Hence, this review was conducted to identify the characteristics and the effects of IcPAB among primary school children. The review protocol was registered on PROSPERO (CRD42021234192). Following the Cochrane guidelines, PubMed, PsycINFO (ProQuest), MEDLINE (EBSCOhost), Embase/Ovid, SportDISCUS (EBSCOhost), Web of Science, Scopus and Academic Search Premier (EBSCOhost) databases were searched to collect data on randomised control trials without a time restriction. The final database search was conducted on the 8 November 2021. Random effects models were used to calculate the effect sizes. The systematic review identified ten eligible studies, nine of which were also included in the meta-analysis. Few studies used the theoretical frameworks and process evaluations. IcPAB showed mixed effectiveness on academic outcomes: i.e., IcPAB had effects on spelling performance (*p* < 0.05) and foreign language learning (*p* < 0.01) but not on mathematics and reading performance. Health behaviors such as moderate-to-vigorous physical activity levels were improved (*p* < 0.01), but IcPAB did not have an impact on cognition outcomes and health outcomes. Given these mixed results, further research is needed underpinned by strong methodological quality, theoretical underpinnings and reliable process evaluation methods.

## 1. Introduction

A large body of evidence shows that academic achievement throughout the early school years is closely associated with health-related behaviors such as children’s physical activities [1]. Integrating physical activity within school curriculum also contributes to reducing the sedentary behaviour [1,2]. Furthermore, evidence suggest that the cognitive function of children in elementary school is associated with physical activity [3], which suggests that children may benefit from classroom-based physical activity [4,5].

Moreover, the cognitive simulation hypothesis suggests that cognitively demanding physical activities would induce significant improvements in cognitive functioning such as problem solving, memorizing, and executive function [2,6,7], which also help in enhancing academic outcomes, such as in mathematics and reading, among primary school children [2,8]. In addition, such physical activities contribute to improving health outcomes such as fitness levels [9], body mass index [10] and anxiety reduction [11] among children.

Yet, despite the benefits of physical activities for primary school students, educational institutes encourage sedentary behaviour in the classroom and force the students to sit most of their time [12,13]. Because the primary goal of an educational institute is to enhance children’s scholastic performance [14], these organizations use conventional teaching methods that encourage more sitting time and sedentary behaviour. As such, decreased levels of physical activity among primary school students have become a significant public health concern [1,15,16,17], as around four out of five primary school students do not meet the internationally recommended physical activity levels [18]. From this, it is clear that activities during physical education classes alone cannot provide adequate opportunities to help students meet the recommended physical activity levels and obtain the benefits of being active [1].

The classroom setting is considered as a good environment to implement physical activity-based interventions [19,20,21,22], as the children are easily reached this way and teachers have a large degree of autonomy to decide the most appropriate time to implement a physical activity break [23,24]. Physical activity breaks within a classroom setting are hypothesized to demonstrate effects on academic performance and cognitive outcomes according to the embodied cognitive loading theory [25] in addition to providing health-related benefits. Hence, studies have attempted to evaluate the relationship of in-classroom physical activity breaks (IcPAB) with academic performance [2,19,26,27,28], cognitive function [2,26,27,29,30,31], health behaviors (e.g., step counts, physical activity levels, sedentary behaviour) [26,32,33], and health outcomes (e.g., physical fitness) [34,35,36,37] regardless of students’ socio-demographic inequalities [19].

These studies, especially randomized controlled trials (RCTs), have provided promising evidence regarding the effects of IcPAB on academic performance, cognition, health behaviors and health outcomes among primary school children [38,39]. It is suggested that a good intervention design should be theory- and evidence-based with robust process evaluation and fidelity mechanisms [40]. However, previous review studies [19,41] have not paid attention to the use of theory and process evaluation/fidelity methods for RCTs, which may limit our understanding of the merit of these interventions. In addition, previous reviews did not include a meta-analysis of intervention effects. Therefore, this study aimed to systematically review and meta-analyze the characteristics, including the theoretical underpinnings and process evaluations, of IcPAB interventions and examine their effectiveness in improving academic performance, cognition, health behaviors, and health outcomes among primary school students.

## 2. Methods

### 2.1. Registration and Protocol

This review was conducted and is reported based on the Preferred Reporting Items for Systematic Reviews and Meta-analyses Protocol (PRISMA-P) guidelines [42,43]. The protocol can be found from the International Prospective Register of Systematic Reviews (PROSPERO) database under CRD42021234192 (https://www.crd.york.ac.uk/prospero/display_record.php?RecodID=234192, accessed on 11 June 2021) and the PRISMA checklist can be found as a Appendix A.

### 2.2. Definitions

Based on previous recommendations pro, which ide definitions for the outcomes being assessed to conserve consistency and clarity throughout the systematic review [41,44,45], the research team first formulated definitions to describe the framework components of the systematic review:

**In-classroom Physical Activity Breaks Interventions:** Any physical activity inside the classroom during regular class time [41] either as short bouts of physical activity performed as a break from academic instruction [41,46] or short bouts of physical activity that include curriculum content [41,47,48] or as an integration of physical activity into lessons in crucial learning areas other than physical education [41,49].

**Academic Outcomes:** learned proficiency in basic skills and content knowledge [50,51,52].

**Cognitive Outcomes:** intelligence, processing speed, and executive function [2,27,53,54,55].

**Health Behaviors:** behaviors that impact health or mortality such as physical activity, sedentary behaviour, exercising, sleeping, adherence to medical guidelines, diet, sexual behaviour, and health-seeking behaviour [56].

**Health Outcomes:** Physical health outcomes such as body mass index (BMI), fitness and diseases [57], and mental and psycho-social outcomes such as anxiety, stress, emotional stability, self-efficacy [58].

**Primary School Students:** Students who belong to the age range from six to thirteen years old [59].

### 2.3. Inclusion Criteria

The research team defined the eligibility criteria (Table 1) based on population, types of interventions, comparisons and types of outcomes that are of interest (PICO):

Publications that were not written in English or included special needs, differently abled and other disadvantaged children, were excluded. In addition, interventions that were carried out both inside and outside the classroom, study protocols and interventions with no data on control groups, studies without original data, and studies that had an age range below six years and above thirteen years were also excluded. Only randomized controlled trials were included in the systematic review; all other study designs were excluded.

### 2.4. Search Strategies

PubMed, PsycINFO (ProQuest), MEDLINE (EBSCOhost), Embase/Ovid, SportDISCUS (EBSCOhost), Web of Science, Scopus and Academic Search Premier (EBSCOhost) databases were searched without a time restriction using the following search keywords and search terms (strings adapted to different databases): “acti* break OR brain break* OR exercise break* OR class* break* OR movement break* OR lesson break* OR bizzi break* OR energi*) AND (primary school OR elementary school) AND (children OR child OR kids OR kid OR adolescents OR adolescent) AND (physical acti* OR exercise OR movement)”, by one author (DP). The final database search was performed on 8 November 2021. Additional hand searches were performed to identify additional papers following a snowball technique by referring to the reference lists of primarily selected papers.

### 2.5. Study Selection

Citations from each database were downloaded into JabRef software, and one author (DP) removed the duplicates. Two authors reviewed the results, DP and YM, first by title and then by abstract. Cochrane’s COVIDENCE online software (Free Version) was used to review the articles. Where ambiguity arose over the title or abstract, DP and YM assessed the study’s eligibility by reading the article’s full text. In the case of doubts, either to include or exclude a study, WL or DY acted as a third assessor to solve such discrepancies.

The initial database search, including five hand-searched articles (Figure 1), provided 2618 publications. After removing 674 duplicates, the team reviewed 1944 papers by title and abstract. Of these,106 articles were retrieved for full-text screening. The reviewers excluded 96 articles that did not meet the inclusion criteria. Ten articles [1,2,5,9,10,11,22,60,61,62] were included in the systematic review, and nine [1,2,5,9,10,11,22,60,61] in the meta-analysis.

### 2.6. Data Extraction

One author D.L.I.H.K.P. extracted data of selected studies for qualitative synthesis. The variables that were extracted were: author, published year, geographical origin, participant characteristics (sample size, age), RCT design (number of study arms, duration, and dosage), theoretical framework and process evaluation methods used, academic outcomes, cognitive outcomes, health behaviour, and health outcomes. The review team categorized the primary studies’ outcomes based on the methods of a previous review study [63]. Extracted data were recorded in an MS Excel Sheet referring to PICO criteria. The data extraction table of the selected full papers was then independently reviewed by two authors Y.D., W.L. Any discrepancies that occurred were cleared through face-to-face discussions by four authors (D.L.I.H.K.P., W.L., Y.D., M.Y.).

### 2.7. Bias Assessment

According to PRISMA-P guidelines, two authors (D.L.I.H.K.P., M.Y.) independently and blindly assessed the risk of bias (RoB) of the studies using the revised Cochrane risk-of-bias tool for randomized trials (RoB 2; [64,65]). RoB 2 version analyzed five domains for individually randomized trials: (1) bias arising from the randomization process, (2) bias due to deviations from intended interventions, (3) bias due to missing outcome data, (4) bias in the measurement of the outcome, and (5) bias in the selection of the reported result including the overall bias score for each study [64]. The result for each study which provided a value of high, low or some concerns, was obtained by responding to the options (Yes, Probably yes, Probably no, No or No information) provided under signaling questions for each risk-of-bias domain. Any disagreement between bias evaluation scores’ risk was resolved through face-to-face discussions, and two authors (W.L., Y.D.) intervened as the tiebreakers where necessary.

### 2.8. Meta-Analysis

When at least two studies were investigating the same broad outcome, with primary data on mean and standard deviation statistics, separate meta-analyses were conducted [66] for the outcome variables (academic outcomes, cognition outcomes, health behaviors, and health outcomes) by comparing pre and post-intervention values or mean differences of each intervention (IcPAB group) and control group. Where there was no baseline data or data on mean differences, the reviewers used post-intervention values (adjusted for baseline differences) given for the intervention and control group of specific studies [19,41].

The meta-analyses were performed using the Review Manager 5.4.1 software (Cochrane, London, UK). When studies reported intervention effects on multiple measures for an outcome, the reviewers included one outcome measure compared with other studies’ outcome measures to prevent duplication of studies under a single outcome [19,41,67]. When there was more than one intervention group in a single study, each intervention group’s result was treated as a separated study [19]. Standardized mean difference (SMD) was used to calculate the effect size of each study by computing the difference between treatment and control means [19,41]. Graphic forest plots with effect estimates with 95% confidence interval were considered for meta-analysis and pooled effect size results. The reviewers used random-effects model as per the guidelines of Cochrane Handbook for Systematic Reviews of Interventions because of the following reasons: (1) the number of investigated studies under each variable ranged from two to seven [66], (2) studies were substantially heterogeneous, and (3) there was a wide variation in health and academic outcomes employed in the different studies. To interpret the pooled effect sizes, Hedges’s *g* with reference to Cohen’s threshold levels: trivial < 0.2, small ≥0.2 to <0.5, moderate ≥0.5 to <0.8, and large ≥ 0.8 [28,68] were used.

To explore the impact of different decisions on meta-analytic results, a sensitivity analysis was performed by excluding or including studies in the meta-analysis based on the methodological quality of the papers, where there were more than two studies in each meta-analysis. If results remained consistent across the different analyses, these were considered robust as they remain identical/similar even after different decisions. It was considered as an indication that the result may need to be interpreted with caution should the results differ after performing the sensitivity analyses. As the meta-analytic review was conducted for continuous outcomes, Egger’s test was performed through JAMOVI 2.0 software to identify the publication bias. Publication bias was detected where *p* < 0.10 [69].

To solve clinical heterogeneity-related problems, the reviewers made sure to make decisions by cross-checking with the PICO criteria and to ensure that all ‘intention-to-treat studies’ were RCTs. In testing the robustness of the matching of the studies for meta-analysis, the statistical heterogeneity was analyzed using graphic forest plots and by calculating the I^2^ statistic (representing the percentage of variance in effect estimates caused by heterogeneity rather than by sampling bias). Threshold level for substantial heterogeneity was set where I^2^ statistic was ≥50% while I^2^ = 0–40% not important, 30–60% moderate heterogeneity, 50–90% substantial heterogeneity, 75–100% considerable heterogeneity [19,67,70]. All the high levels of I^2^ were reported with caution [19,41,67] as the number of meta-analyzed studies were less than ten under each outcome.

## 3. Results

### 3.1. Study Characteristics

The characteristics of each study are summarized in Table 2. Overall, the studies were published from 2013 to 2021 and conducted in Australia [62,71]; Ireland [1,10,61]; Netherlands [9]; Switzerland [2,11]; and United States [5,22], which are all western and high income countries. The sample size varied from 40 [5] to 467 children [9]. Across included studies, participant ages ranged from seven [2] to twelve [1,9,60] years old. All the studies consisted of both male and female participants. None of the studies analyzed the effects of IcPAB on the ethnicity of the students, although Layne and colleagues mentioned that their sample consisted of African American students [5]. However, only one study had stratified its outcomes by gender [62]. The effectiveness of the IcPAB was measured after either implementing a two-arm [1,5,9,10,22,60,61,62] or three-arm [2,11] RCT. Four out of ten RCTs were cluster randomized controlled trials (C-RCTs) [1,5,62,72]. The duration of the C-RCTs ranged from four [5] to nine [9] weeks, while RCT interventions’ span ranged from a day [60] to eight months [22]. Two IcPABs were implemented for less than a week [60,61], four were implemented between two and six weeks [1,5,62,73] and three interventions were implemented for more than 12 weeks [2,10,22]. With the exception of two studies (five minutes for an activity break) [10,62], 80% of the interventions allocated ≥ 10 min for an IcPAB session. The total provision for an IcPAB per day ranged from 15 to 20 min [22]. In terms of the total intensity of IcPAB intervention, three studies were between 10 and 50 min [1,60,61], four studies were between 140 and 630 min [5,9,11,62], and three studies were between 1260 and 4800 min [2,10,22].

#### 3.1.1. Theoretical Foundations and Process Evaluation Methods

Two studies (20% of total selected studies) included theoretical support such as the ecological model [62], social cognitive theory [62], behaviour change theory [1] and the B-COM model encompassed in behaviour wheel change framework [1,62]. Other studies did not report a theoretical underpinning for the intervention or a rationale behind the IcPAB activities. Five studies reported the fidelity and process evaluation mechanisms for the studies and used self-evaluated questionnaires by facilitators [1], self-completed daily logs for intensity and accuracy of IcPAB by facilitators [2,11,22,62], post-intervention discussions [62], awarding intensives such as memberships for successfully following intervention guidelines [22], and utilizing drop-in observation visits by student researchers [22]. However, in addition to reporting fidelity mechanisms, only two studies [2,9] discussed the teachers’ compliance with the implementation of IcPAB. No studies addressed the students’ compliance in attending the IcPAB.

#### 3.1.2. Intervention Content

According to the information provided in Table 2, IcPAB used in each intervention varied from teacher-led physical exercises [1,2,10,11,22,60,61] to video-based activities [5,62,72]. Some studies seemed to be curriculum-linked, [11] while some were used as brain breaks or cognitively challenging activities [2].

#### 3.1.3. Outcomes Evaluated

Six out of ten of the studies examined the effects of IcPAB on primary school children’s academic performance [2,5,11,22,60,62]. Mathematics [2,5,22,60,62], reading [2,22,62], spelling skills [2] and foreign language learning [11] achievements were the academic outcomes considered in the RCT-based IcPAB interventions.

Cognition outcomes included executive functions such as inhibition [2,5,9], updating [2], shifting [2], attention performance [9,11], fluid intelligence [22], reaction time [5], semantic memory retrieval [9], perceived task difficulty [60] and mental effort [60].

Moderate-to-vigorous physical activity (MVPA) [1,9,10,62], step count [1,22] and sedentary behaviour [1,9] were identified as the health behaviors, which were quantitatively measured using accelerometers [1,2,62] and pedometers [22,61]. Some studies [2,11] used these behavioral outcomes either as manipulation check variables or baseline data through qualitatively [2] or quantitatively [11] measured physical activity levels of children.

Aerobic fitness [9], BMI [10], skinfold measurement changes [10], and test-anxiety [11] were assessed as health outcomes. However, some of the studies measured health outcomes such as aerobic fitness [2], BMI [2,9,11,61], or gross motor coordination [2] as descriptive variables of the participants, not as the intervention effects.

### 3.2. Bias Assessments

Risk of bias assessment indicated to some concerns over the methodological quality among seven of the studies. Two studies were assessed as having a high risk of bias [5,11] and one study was assessed as having a low risk of bias [2] (Table 3).

### 3.3. Intervention Effects

Effectiveness of the IcPAB interventions on academic outcomes (*n* = 13 intervention samples from 5 studies [2,5,11,22,60]), cognitive outcomes (*n* = 12 intervention samples from 4 studies [2,5,9,11]), health behavior (*n* = 5 studies [1,9,10,22,61]), and health outcomes (*n* = 2 intervention samples from a single study [60]) were quantitatively analyzed (Table 4).

#### 3.3.1. Intervention Effects on Academic Performance

Six intervention samples (*n* = 725 students), were meta-analyzed from four RCTs [2,5,22,60] for mathematics achievement (Figure 2).

The quantitative synthesis indicated trivial to small effects (SMD = 0.15, 95% CI [−0.13 to 0.43], *p* = 0.30) favoring the mathematics performance (Panel 1 in Figure 2) of intervention groups (I^2^ = 59%, *p* = 0.03; Egger’s regression = −0.530, *p* = 0.624; (Table 4)). The sensitivity analysis confirmed that finding (SMD = 0.35, 95% CI [−0.15 to 0.86], *p* = 0.17; I^2^ = 77%, *p* = 0.01; Egger’s regression = −1.686, *p* = 0.190) with five intervention samples (*n* = 632 students). Meta-analysis for reading (based on three samples from two studies [2,22]; *n* = 617 students) indicated a trivial to small pooled effect size that was not significant favoring the control group before (SMD = −0.07, 95% CI [−0.25 to 0.11], *p* = 0.43; I^2^ = 11%, *p* = 0.33; Egger’s regression = −1.624, *p* = 0.351; Panel 2) and after the sensitivity analysis (Panel 2 in Figure 2).

Effectiveness of the IcPAB interventions on spelling skills (Panel 3 in Figure 2) was evaluated through a single study [2] using two different intervention arms (*n* = 188 students). The result with a statistically significant, large, pooled effect estimate (SMD = 2.13, 95% CI [0.21 to 4.05], *p* = 0.03; I^2^ = 11%, *p* = 0.45) confirmed that the classroom-based physical activity breaks might have an impact on primary school kids’ spelling performance. Two intervention samples (*n* = 137 students) from another single study [11] confirmed the effectiveness of IcPAB in improving the foreign language learning ability of students. The analysis reported a statistically significant, moderate to large effect size (SMD = 0.80, 95% CI [0.21 to 1.39], *p* = 0.008) for foreign language learning (Panel 4 in Figure 2) with a moderate to substantial amount of heterogeneity (I^2^ = 64%, *p* = 0.09).

#### 3.3.2. Intervention Effects on Cognition

Four intervention samples from three studies [2,5,9] were meta-analyzed for the effects of inhibition among primary school children. With a significant level of considerable heterogeneity (I^2^ = 97%, *p* < 0.00001), it was found that the IcPABs do not have significant effects on the inhibitory performance (SMD = −0.64, 95% CI [−1.85 to 0.56], *p* = 0.30) among the participants (Panel 1A in Figure 3). After performing the sensitivity analysis, it was confirmed that the inhibition performance was not improved by the intervention with a significant moderate to large, pooled effect size (SMD = −1.48, 95% CI [−2.33 to −0.64], *p* = 0.0006), while there was a considerable heterogeneity level (I^2^ = 83%, *p* = 0.01) without publication bias (Egger’s regression = 0.417, *p* = 0.717; (Panel 1B in Figure 3).

Two intervention samples from the same study [2] showed no significant trivial to small pooled effect sizes favoring the control group (SMD = −0.07, 95% CI [−0.38 to 0.24], *p* = 0.65) with a less important heterogeneity level (I^2^ = 15%, *p* = 0.28) for updating (Panel 2 in Figure 3).

However, the same study [2] reported that the classroom physical activity breaks may have impacts on the shifting performance (SMD = 0.15, 95% CI [−0.14 to 0.44], *p* = 0.31; I^2^ = 0%, *p* = 0.42) of the executive function with moderate to large effects (Panel 3 in Figure 3).

Three intervention samples from two studies showed positive impacts of IcPAB on children’s attention performance [9,11] with small to large pooled effects (SMD = 0.31, 95% CI [−1.15 to 1.77], *p* = 0.67; I^2^ = 98%, *p* < 0.00001; (Panel 4 in Figure 3).

#### 3.3.3. Intervention Effects on Health Behaviour

Physical activity breaks, which were implemented inside the classroom [1,9,10] indicated a large significant pooled effect size (*n* = 605 students; SMD = 3.55, 95% CI [3.29 to 3.81], *p* < 0.00001) favoring the intervention groups for their improved MVPA levels (Panel 1 in Figure 4) (I^2^ = 97%, *p* < 0.00001; Egger’s regression = 2.546, *p* = 0.238). Meta-analysis of the step count (Panel 2 in Figure 4) indicated a trivial to small pooled effects favoring the IcPAB interventions (SMD = 0.15, 95% CI [−0.19 to 0.50], *p* = 0.39; I^2^ = 57%, *p* = 0.13) based on a sample size of 516 primary school students [22,61].

Two studies [1,9] with 498 elementary level students reported that the sedentary time within a classroom setting was reduced by IcPAB interventions (Panel 3 in Figure 4) as the quantitative synthesis reported moderate to large pooled effects (SMD = 1.10, 95% IC [−1.19 to 3.39], *p* = 0.35) favoring the intervention groups (I^2^ = 99%, *p* < 0.00001).

#### 3.3.4. Intervention Effects on Health Outcomes

Two intervention samples (*n* = 68 students) from a single study [60], which was meta analyzed for test-anxiety as a mental health outcome provided a moderate to large pooled effect size (SMD = 0.16, 95% CI [−0.31 to 0.64], *p* = 0.50; I^2^ = 0%, *p* = 0.71) favoring the IcPAB intervention (Figure 5).

## 4. Discussion

### 4.1. Sample, Intervention Characteristics, Outcomes, Theory, and Process Evalaution

Data from 1538 primary school students (from seven to twelve years old) in 10 studies were analyzed to assess the characteristics of IcPAB-related interventions and evaluate effectiveness in improving academic performance, cognition, health behaviors, and health outcomes. Interestingly, all the studies were conducted in high income countries, and only one study reported whether the gender of the students would influence the results of the IcPAB studies. Effects of IcPAB by ethnicity could not be found. Such weaknesses not only illustrate the need for more studies in this area, but also indicates a need for studies in low- and middle-income countries. Only four of the studies used C-RCT designs [1,5,62,72], even though C-RCTs are recommended to evaluate the interventions effects of clusters such as classrooms [74]. Hence, C-RCT designs are encouraged for future IcPAB interventions. The intervention duration was less than 12 weeks in the majority of studies [1,5,9,60,61,62,73], and allocated ≥10 min per an activity break, which is consistent with previous research findings [41,75]. The total intensity of intervention varied significantly, ranging from 10 min to 4800 min. The suitable IcPAB intervention intensity (dosage) needs to be identified in the future.

Most studies (*n* = 8) demonstrated average methodological quality, with concerns around the randomization procedure, handling of missing data and the outcome evaluation. When the risk of bias for methodological quality is relatively high in RCTs, the results should be interpreted with caution [19,41]. Our results suggest that the methodological quality of RCTs examining in-classroom physical activity breaks should be improved in future studies [19,41].

Most of the studies focused on understanding the effects of IcPAB on academic achievements, cognition health behaviour and health outcomes. This may be because there are theoretical assumptions and evidence for the relationships between such outcomes [2,28,35,76,77] even though those are under researched among the primary school level children [5,41]. However, none of the studies focused on diet and intake of vitamin supplements, which are important aspects of child growth and education [78]. This emphasises the need for future studies that examine the contribution of diet on IcPAB intervention among elementary school children.

This study also suggests the need for theoretical frameworks [13,79] in designing IcPAB interventions, with well explained process evaluation and fidelity methods [40,80,81]. The use of self-completed daily logs [2,11,22,62] for intensity and accuracy of IcPAB intervention delivery by facilitators seemed to be a popular method for fidelity and process evaluation.

### 4.2. Effectiveness

Previous review studies [13,19,27,35,41,75,82,83], focused on physical activity breaks that were conducted both inside and outside the classrooms without limiting the focus to RCT designs. These studies found that the physical activity breaks can have mixed effects on academic performance. In line with these study findings, current analysis also identified that the IcPAB have mixed effects on academic performance. The reasons for having mixed results for academic achievement could be due to quality, evaluation content, and the standardization of the test for each academic outcome [19,41]. The embodied cognitive load theory suggests that the intensity, load and the extent of physical activity integration into the curriculum affects the academic performance of a student [25]. The type of IcPAB (curriculum-based or general physical activity breaks) and its duration might moderate the effects of IcPAB on academic performance [41,75]. Therefore, more studies should be conducted to identify the accurate effects on academic performance by comparing different types of IcPAB among primary school students [27,41]. Based on the results for executive functions such as inhibition, updating and shifting, as well as the results for mathematics and reading, it can also be assumed that there is a positive association between executive functions and the academic performance of children [8,26,84,85,86].

Current findings suggest that the effects of IcPAB on cognitive function of primary school children are inconsistent and mixed. Previous reviews [19,41,75,87], which focused on school-based studies that incorporated physical activity breaks both in and outside the classroom and included child populations without an age restriction, reported similar results. Therefore, it can be suggested that the venue (in-classroom or outside the classroom) does not play a crucial role within the school setting in improving cognition through physical activities. According to the cognitive simulation hypothesis, the cognitive demand levels of the physical activity influence the improvements of the cognition [2,6,7]. Hence, it is possible that the IcPAB was not cognitively demanding enough, given that most of the subdomains of the cognition did not have intervention effects. In addition, as explained in Watson’s and Masini’s studies [19,41], the intensity of activity breaks, the validity and the reliability of the measurements used to evaluate the cognitive performance, the smaller sample sizes, and the inconsistency of the most appropriate amount of physical activity breaks for a cognitive arousal have likely contributed to the conflicting results for cognitive performance among children.

Referring to the effects of classroom-based physical activity breaks on health behaviour, it was found that the MVPA levels of the elementary level students improved. Hence, in line with Masini’s meta-analysis [19], but contradictory to another meta-analysis [41], this review confirmed the positive effects of IcPAB for improving physical activity levels. Yet, it should be noted that both those meta-analyses [19,41] were generated by referring to all types of study designs in contrast to the current review which was restricted to RCTs only. However, step count and sedentary behaviour did not indicate pooled effect estimates favoring the IcPAB interventions, contradictory to Masini’s findings [19]. Therefore, similar to a previous recommendation [41], this study also suggests that the results on health-related behaviors be interpreted with caution due to the small number of studies (*n* = less than three studies) included in the meta-analysis.

Finally, less than two studies were identified that studied the IcPAB’s effects on health outcomes such as aerobic fitness and BMI. Even though a systematic review [13] reported positive effects on BMI contrary to the current finding from the qualitative synthesis, further studies are warranted to measure pooled effects before providing a conclusive result. Only the effects on test-anxiety as a mental health-related outcome could be analyzed in this review. Even though, the test-anxiety did not provide statistically significant results, it should not be generalized, as the effects sizes were generated from a small sample size based on a single three-arm RCT.

### 4.3. Limitations and Recommendations

There are several limitations of this review. Identifying eligible studies was limited to English-language publications. The study did not analyze the effects of IcPAB on children with special education needs. As all the studies were published with data from high income countries, the outcomes cannot be generalized to include the entire world. In terms of the outcomes, there were seven or less studies under each outcome. Hence, the smaller number of studies limited the possibilities of conducting sub-group analysis in the quantitative synthesis. In addition, 90% of the studies did not analyze the long-term effects of the intervention as they abstained from the follow-up stage [1,2,5,9,10,11,22,61]. Notably, none of the findings indicated publication bias except for in attention performance. However, there were considerable levels of heterogeneity for some outcomes, as well as concerns related to the risk of bias. This limits the interpretation of the current study, as less rigorous studies might be biased toward overestimating or underestimating the intervention effects.

Yet, despite these limitations this review clearly emphasized the existing gaps in classroom-based physical activity break interventions. This demonstrates that further rigorous and well-designed IcPAB programs are needed to enhance the intervention effects on elementary students’ academic performance, cognition, health behaviors and health outcomes. In particular, theoretical underpinnings such as the COM-B behaviour model [1] can be integrated to these intervention designs to obtain positive results [13,35,40]. The COM-B model proposes that people need capability (C), opportunity (O) and motivation (M) to perform a particular behaviour (B) [1]. There are two studies [38,88] using the COM-B model to identify the capabilities, opportunities and motivations of the IcPAB facilitators (e.g., teachers). Based on these findings, the authors then applied certain behaviour change techniques and intervention functions such as education, training, and enablement to improve intervention effects by ensuring the fidelity of their trials. In addition, many studies claimed that there are difficulties for classroom teachers in implementing physical activity breaks [5,23,89] due to high curriculum demands. Pure educational time might also be shortened due to the implementation of IcPAB. Therefore, it would be promising to use curriculum related IcPAB in response to teacher concerns over tight curriculum and insufficient education time. Furthermore, policy level recommendations for teachers from the education authorities to implement compulsory daily IcPAB during lessons are also needed to promote activity breaks for improving the academic performance, cognition, and health outcomes of elementary level students. In addition, it was found that the majority of the studies in the review did not analyze the compliance rate on IcPAB both for teachers and students. Therefore, compliance issues should be taken into consideration in the future. Furthermore, total intensity of intervention may be correlated with the intervention effects on academic performance and health outcomes [75,90]. Therefore, moderation analyses of the IcPAB intervention intensity should be warranted in future meta-analyses. Furthermore, the effects of IcPAB on differently abled and children with special educational needs are suggested to be addressed in the future.

## 5. Conclusions

Our study demonstrated mixed effectiveness of IcPAB on academic outcomes (IcPAB had positive effects on spelling performance and foreign language learning but not on mathematics and reading performance) and health behaviors (moderate-to-vigorous physical activity levels were improved), but IcPAB did not have an impact on cognition outcomes and health outcomes. Moreover, few studies used theoretical frameworks and process evaluations. Importantly, our study generally included few studies examining the same outcomes, indicating that the effects of IcPAB interventions are under-researched, especially in relation to gender, low- and middle-income countries and the Asian region. A practical-knowledge gap was also found, as the time allocation for IcPAB sessions seemed to differ from what the classroom teachers desired. This study emphasizes the need for improved methodological quality of the RCT designs, specifically in relation to randomization and blinding process, missing data handling and the outcome evaluation. Finally, this study demonstrates that more classroom-based physical activity break intervention studies with RCT designs are required for primary school children to generalize the current findings on academic achievement, cognition, health behaviors and health outcomes.

## Figures and Tables

**Figure 1 ijerph-19-09479-f001:**
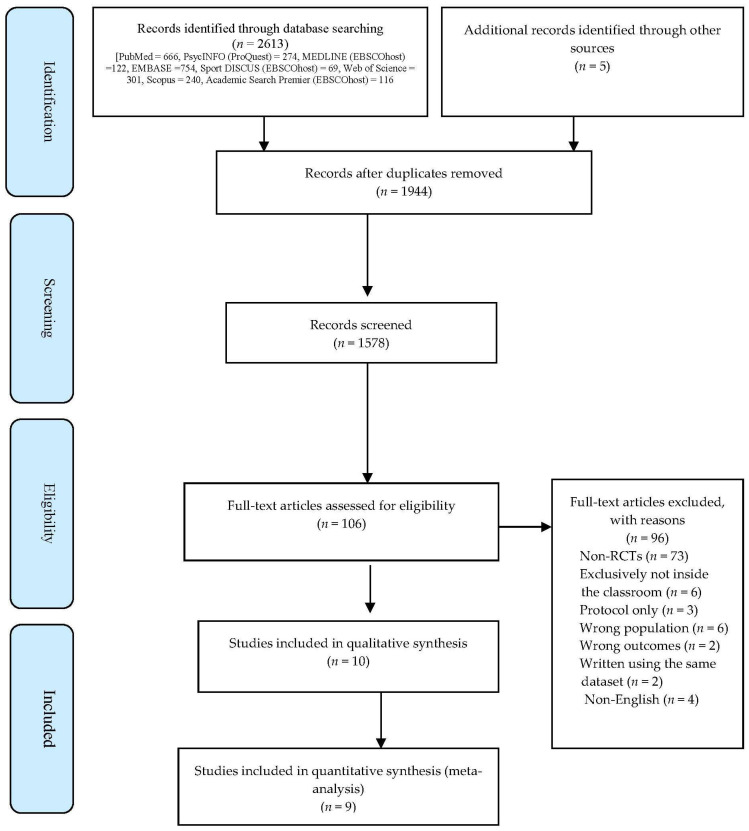
PRIMA Flowchart for selecting studies.

**Figure 2 ijerph-19-09479-f002:**
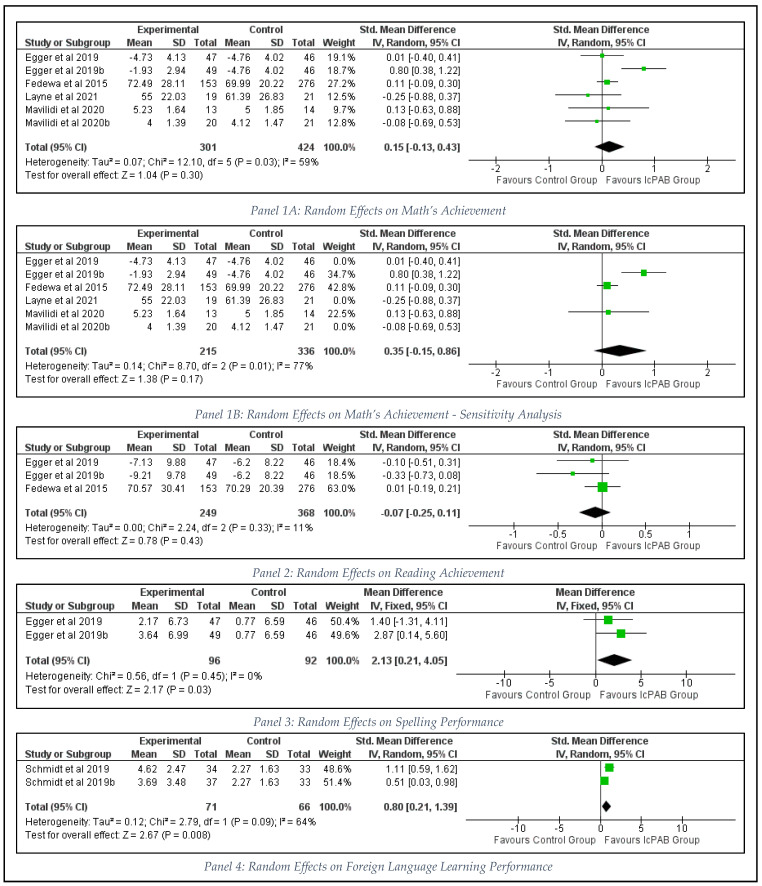
Effects of IcPAB on Academic Achievement Outcomes.

**Figure 3 ijerph-19-09479-f003:**
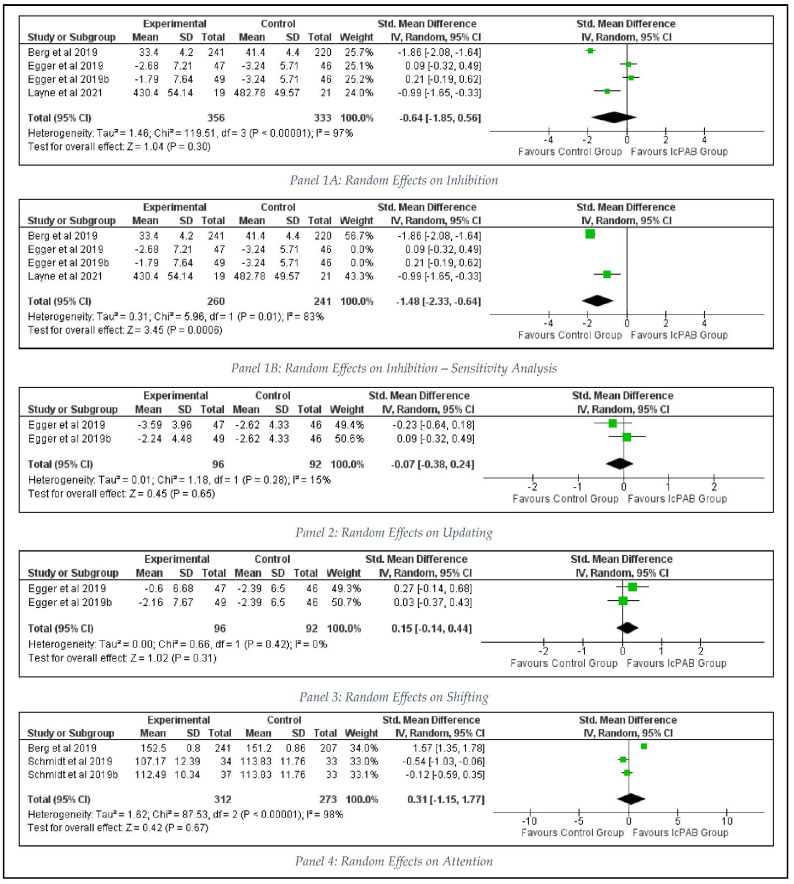
Effects of IcPAB on Cognitive Outcomes.

**Figure 4 ijerph-19-09479-f004:**
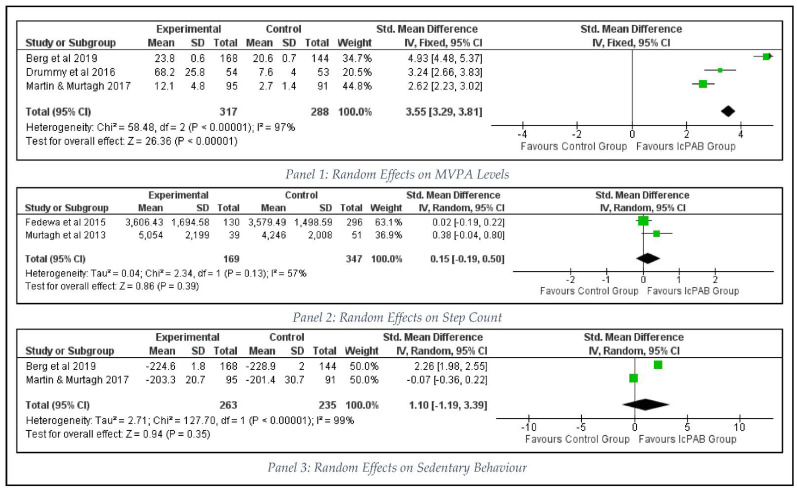
Effects of IcPAB on Health Behaviour Outcomes.

**Figure 5 ijerph-19-09479-f005:**
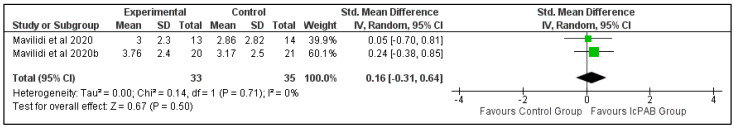
Effects of IcPAB on Test Anxiety (health-related outcome).

**Table 1 ijerph-19-09479-t001:** Criteria for Eligible Studies.

Criteria	Inclusion
**P**opulation (Participants)	Children obtaining primary education in schools (6–13 years old)
Types of **I**nterventions	Activity breaks interventions carried out inside the classroom with original primary data
**C**omparators (Comparisons)	Intervention vs. control in randomized controlled trails
Types of **O**utcomes that are of interest	Academic outcomes, cognitive outcomes, health behaviors, and health outcomes

**Table 2 ijerph-19-09479-t002:** Summary of selected studies (k = 10).

Author ID (Year), Country	Characteristics of the Participants	Intervention Characteristics				Evaluated Outcomes (Measuring Points for Outcomes)	Main Results [M(SD) or ŷ (SE) or B (CI)]
	Sample Size (*n*); Age (Mean, SD)	RCT Design (Number of Study Arms); Duration of IcPAB; Intensity per Day; Total Intensity of Intervention	Theoretical Frameworks	Fidelity and Process Evaluation Methods	Intervention Content	Academic Outcomes, Cognitive Outcomes, Health Behaviour Outcomes, and Health Outcomes	
Murtagh et al. (2013) [61] Ireland	*n* = 90 (IG = 39; CG = 51); Age: 9.3 (1.4)	RCT (2-arm); 5 days; 10 min per day in one session; 50 min	None	Record sheets for students Compliance rate: None for teachers and students	Techers of the intervention classes led 10 min IcPAB for five consecutive days. A series of mobility, stretching and pulse-raising exercises performed to music beside students’ desks. The activities were summarised to teachers through a poster, teacher notes, and a music CD by a researcher.	2 measuring points (baseline and follow-up) Academic outcomes: Nil. Cognitive outcomes: Nil. Health behaviour outcomes: In-school step counts (Pedometer). Health outcomes: Nil.	**Step count**—IG: M = 5054 (SD = 2199); CG: M = 4246 (SD = 2008)
Fedewa at al (2015) [22] USA	*n* = 460 (IG = 156; CG = 304), Age: Grade 3 to 5	RCT (2-arm); 8 months; 20 min per day in one session; 4800 min	None	Record sheets for students and gym memberships for teachers through a lucky draw Compliance rate: None for teachers and students	Teachers led integrated PA into the core-curricular five days a week using standardised movement cards. The cards consisted of aerobic-based activities such as jumping jacks or finding different decks of cards that are spread around the class.	2 measuring points (pre-test during fall and post-test during spring) Academic outcomes: Mathematics; Reading. Cognitive outcomes: Fluid intelligence. Health behaviour outcomes: in-school step counts (pedometer). Health outcomes: Nil.	**Mathematics**—IG: ŷ = 72.49 (SE = 28.11); CG: ŷ = 69.99 (SE = 20.22); **Reading**—IG: IG: ŷ = 70.57 (SE = 30.41); CG: ŷ = 70.29 (SE = 20.39); **Fluid intelligence**—IG: ŷ = 0.62 (SE = 0.55); CG: ŷ = 0.01 (SE 0.07) **Step count**—IG: ŷ = 3606.43 (SE = 1694.58); ŷ = CG: 3579.49 (SE = 1498.59);
Egger et al. (2019) [2] Switzerland	*n* = 142 (IG_1_ = 47; IG_2_ = 49; CG = 46), Age: 7.82 to 7.944 (0.41 to 0.40)	RCT (3-arm); 20 weeks; 20 min per day in two 10-min sessions; 2800 min	None	Record sheets for teachers Compliance rate: Teachers: implemented 145.4 activities out of 200 Students: None	IG_1_ performed specific PA which would challenge the EFs such as “horserace” games with cognitive demands. IG_2_ performed aerobic PA such as “horserace” games without cognitive demands. CG performed fine motor tasks without any physical exertion when sitting.	2 measuring points (pre-test and post-test) Academic outcomes: Mathematics; Reading; Spelling Cognitive outcomes: EFs-Updating, Inhibition, Shifting Health outcomes: Nil.	**Mathematics**—IG_1_: M = −4.73 (SD = 4.13), IG_2_: M = −1.93 (SD = 2.94), CG: M = −4.76 (SD = 4.02); **Reading**—IG_1_: M = −7.13 (SD = 9.88), IG_2_: M = −9.21 (SD = 9.78); CG: M = −6.2 (SD = 8.22); **Spelling**—IG_1_: M = 2.17 (SD = 6.73), IG_2_: M = 3.64 (SD = 6.99), CG: M = 0.77 (SD = 6.59); **Updating**—IG_1_: M = −3.59 (SD = 3.96), IG_2_: M = −2.24 (SD = 4.48), CG: M = −2.62 (SD = 4.33); **Inhibition**—IG_1_: M = −2.68 (SD = 7.21), IG_2_: IG_2_: M = −1.79 (SD = 7.64), CG: M = −3.24 (SD = 5.71); **Shifting**—IG_1_: M = −0.6 (SD = 6.68), IG_2_: M = −2.16 (SD = 7.67), CG: M = −2.39 (SD = 6.5)
Berg et al. (2019) [72] Netherlands	*n* = 448 to 467 (IG 239 to 244; CG: 207 to 223), Age: 10.9 (0.7)	C-RCT (2-arm); 9 weeks; 10 min per day in one session; 630 min	None	None Compliance rate: Teachers: 4.4 IcPAB per week Students: None	Three videos freely available from Ubisoft were used via YouTube. Children had to mimic the figure in the video. The videos had acute PA intercity.	2 measuring points (pre-test and post-test) Academic outcomes: Nil. Cognitive outcomes: Attention; Inhibition; Semantic memory retrieval. Health behaviour outcomes: MVPA and SB (accelerometer). Health outcomes: Aerobic fitness.	**Attention**—IG: M = 152.5 (SD = 0.8), CG: M = 151.2 (SD = 0.86); **Inhibition**—IG: M= 33.4 (SD = 4.2), CG: M = 41.4 (SD = 4.4); **Semantic memory retrieval**—IG: M = 11.7 (SD = 0.16), CG: M = 11.9 (SD = 0.17); **MVPA**—IG: M = 23.8 (SD = 0.6); CG: M = 20.6 (SD = 0.7); **SB**—IG: M = −224.6 (SD = 1.8); CG: M = −228.9 (SD = 2); **Aerobic fitness**—IG: M = 48.9 (SD = 0.2); CG: IG: 48.8 (SD:0.2)
Schmidt et al. (2019) [11] Switzerland	*n* = 104 (IG_1_ = 34; IG_2_ = 37; CG = 33), Age: 9.04 (0.70)	RCT (3-arm); 2 weeks; 10 min per day in one session; 140 min	None	None Compliance rate: None for teachers and students	French words for animal names were showed on a big screen with pictures and audio. IG_1_ children had to enact the movements in a video indicated by the animal names to be learned. IG_2_ children had to do the same as IG_1_ when running on the spot.	2 measuring points (pre-test and post-test) Academic outcomes: Foreign language learning. Cognitive outcomes: Attention. Health behaviour outcomes: Nil. Health outcomes: Nil.	**Foreign language learning**—IG_1_: M = 4.62 (SD = 2.47); IG_2_: M = 3.69 (SD = 3.48); CG: M = 2.27 (SD = 1.63); **Attention**—IG_1_: M = 107.17 (SD = 12.39); IG_2_: M = 112.49 (SD = 10.34); CG: M = 113.83 (SD = 11.76);
Watson et al. (2019) [62] Australia	*n* = 341 (IG = 123; CG = 218), Age: 9.22 (0.61) to 9.07 (0.63)	C-RCT (2-arm); 6 weeks; 15 min per day in 5 min three sessions; 630 min	COM-B model; SCT; EM	Ratings for IcPAB by students; Focus group discussions with teachers and students Compliance rate: None for teachers and students	Teachers implemented MVPA such as drama games. i.e., “students move around the classroom as the music plays. When the music stops, the teacher calls out a body part and the students return to their chair and place the selected body part on their chair”.	2 measuring points (pre-test and post-test) Academic outcomes: Mathematics; Reading; On-task behaviour. Cognitive outcomes: Nil. Health behaviour outcomes: School-day MVPA (accelerometer). Health outcomes: Nil.	**Mathematics**—B = 1.86 (95% CI: −0.01, 3.73) **Reading**—B = −0.31 (95% CI: (−8.08, 7.81) **On-task behaviour**—B = 16.17 (95% CI: 6.58, 25.76) **MVPA**—B = 1.26 (95% CI: −3.78, 6.30)
Layne et al. (2021) [5] USA	*n* = 40 (IG = 19; CG = 21), Age: 8 to 9 years	C-RCT (2-arm); 4 weeks; 10 min per day in one session; 280 min	None	None Compliance rate: None for teachers and students	Students had to play FitNexx 1.0 active video game everyday at the school before their mathematics lesson. The game had movement based MVPA with fun elements.	2 measuring points (pre-test and post-test) Academic outcomes: Mathematics; Cognitive outcomes: Inhibition; Reaction time. Health behaviour outcomes: Nil. Health outcomes: Nil.	**Mathematics**—M = 55 (SD = 22.03); CG: M = 61.39 (SD = 26.83) **Inhibition**—M = 430.4 (SD54.14); CG: M = 482.78 (SD = 49.57) **Rection time**—M = 430.4 (54.14); CG: M = 492.73 (SD = 57.98)
Martin & Murtagh (2017) [1]	*n* = 186 (IG = 95; CG = 91), Age: 9.1 (0.9)	C-RCT (2-arm); 5 days; 10 min per day in one session; 50 min	BCW + COM-B; BCT	Record sheets and questionnaires for teachers Compliance rate: None for teachers and students	Techers delivered curriculum-related physically active lessons during English and Mathematics lessons. The PA could be modified by the teachers to fit with their schedules.	2 measuring points (pre-test and post-test) Academic outcomes: Nil. Cognitive outcomes: Nil. Health behaviour outcomes: School-day MVPA; SB (accelerometer). Health outcomes: Nil.	**MVPA**—IG: M = 12.1 (SD = 4.8); CG: M = 2.7 (SD = 1.4) **SB**—IG: M = 203.3 (SD = 20.7); CG: M= −201.4 (SD = 30.7)
Mavilidi et al. (2020) [60] Australia	*n* = 68 (IG = 33; CG = 35), Age: 11 to 12 years	RCT (2-arm); 1 day; 10 min per day in one session; 10 min	None	None Compliance rate: None for teachers and students	Students were asked to do PA such as push-ups, star jumps, penguin movements, burpees, and running on the spot. Other IG played the hangman game on the school’s whiteboard.	3 measuring points (pre-test, during the test and post-test) Academic outcomes: Mathematics. Cognitive outcomes: Invested mental effort; perceive task difficulty. Health behaviour outcomes: Nil. Health outcomes: Test-anxiety	**Mathematics**—IG_1_: M = 5.23 (SD = 1.64); IG_2_: M = 4.00 (SD = 1.39); CG: M = 5 and M = 4.12 (SD = 1.85 and SD = 1.47) **Invested mental effort**—IG_1_: M = 4.35 (SD = 0.98); IG_2_: M = 5.28 (SD = 1.29); CG:M= 5.03 and M = 4.62 (SD = 1.44 and SD = 1.75) **Perceive task difficulty**—IG_1_: M = 5.12(SD = 1.45); IG_2_: M = 5.22 (SD = 1.56); CG:M = 4.78 and M = 4.78 (SD = 1.75 and SD = 1.57) **Test-anxiety**—IG_1_: M = 3.00 (SD = 2.3); IG_2_: M = 3.76 (SD = 2.4); CG: M = 2.86 and M = 3.17 (SD = 2.82 and SD = 2.5)
Drummy et al. (2016) [10] Ireland	*n* = 107 (IG = 54; CG = 53), Age: M = 9.5	RCT (2-arm); 12 weeks; 15 min per day in 5 min three sessions; 1260 min	None	None Compliance rate: None for teachers and students	Students performed PA chosen by the teachers from an activity pack with 40 exercises. PA started with gentle jogging on the spot as a warmup for less than 1min, followed by MVPA such as hopping, jumping, and running on the spot, and scissor kicks.	2 measuring points (pre-test and post-test) Academic outcomes: Nil. Cognitive outcomes: Nil. Health behaviour outcomes: MVPA (accelerometer). Health outcomes: BMI; Skinfold measures.	**MVPA**—IG: M = 68.2 (SD = 25.8); CG: M = 7.6 (SD = 4.00) **BMI**—IG: M = 19.3 (SD = 3.3); CG: M = 18.3 (SD = 2.6) **Skinfolds**—M = 41.2 (SD = 7.1); CG: M = 40.1 (SD = 6.2)

*Note*. RCT: randomized controlled trial; C-RCT: cluster randomized controlled trial; IG: Intervention group; CG: Control group; M: mean; SD: standard deviation; ŷ: estimated value based on regression; B: unstandardized beta; CI: confidence interval; IcPAB: in-class physical activity breaks; PA: Physical activity; MVPA: moderate-to-vigorous physical activity; SB: sedentary behaviour; BMI: body mass index; BCW: Behaviour change wheel; COM-B: Capability, opportunity and motivation model; SCT: Social cognitive theory; EM: Ecological model; BCT: behaviour change theory.

**Table 3 ijerph-19-09479-t003:** Risk of Bias in the Selected Studies.

Study ID	D1	D2	D3	D4	D5	Overall	Key	
Berg et al 2019 [72]								**High Risk**
Egger et al 2019 [2]								**Low Risk**
Fedewa et al (2015) [22]								**Some Concerns**
Layne et al 2021 [5]							**D1**: Randomisation process **D2**: Deviations from the intended interventions **D3**: Missing outcome data **D4**: Measurement of the outcome **D5**: Selection of the reported result **Overall**: Overall risk of bias	
Murtagh et al 2013 [61]								
Schmidt et al 2019 [11]								
Watson et al 2019 [62]								
Martin et al 2017 [1]								
Mavilidi et al 2020 [60]								
Drummy et al 2016 [10]								

**Table 4 ijerph-19-09479-t004:** Meta-analysis: IcPAB’s effects on the academic achievement, cognition, health behaviors, and health outcomes.

Outcome Variable	*k*^1^ (*n*) ^2^	Effect Estimate (SMD) ^3^ [95% CI]	Significance of Effect Estimates/*p*	Heterogeneity Statistics	Egger’s Regression	Direction Towards
**Academic Achievement**						
Mathematics	6 (725)	0.15 [−0.13, 0.43]	*p* = 0.30	I^2^ = 59% (*p* = 0.03)	−0.530 (*p* = 0.624)	IcPAB Group
Reading	3 (617)	−0.07 [−0.25, 0.11]	*p* = 0.43	I^2^ = 11% (*p* = 0.33)	−1.624 (*p* = 0.351)	Control group
Spelling	2 (188)	2.13 [0.21, 4.05]	***p* = 0.03**	I^2^ = 0% (*p* = 0.45)	N/A	IcPAB Group
Foreign Language	2 (137)	0.80 [0.21, 1.39]	***p* = 0.008**	I^2^ = 64% (*p* = 0.09)	N/A	Control group
**Cognition**						
Inhibition	4 (689); 2 (501) ^APS^	−1.48 [−2.33, −0.64] ^APS^	***p*****= 0.0006** ^APS^	I^2^ = 83% (*p* = 0.01)	0.417 (*p* = 0.717)	Control group
Updating	2 (188)	−0.07 [−0.38, 0.24]	*p* = 0.65	I^2^ = 15% (*p* = 0.28)	N/A	Control group
Shifting	2 (188)	0.15 [−0.14, 0.44]	*p* = 0.31	I^2^ = 0% (*p* = 0.42)	N/A	IcPAB Group
Attention	3 (585)	0.31 [−1.15, 1.77]	*p* = 0.67	I^2^ = 98% (*p* < 0.00001)	−10.875 (*p* = 0.058)	IcPAB Group
**Health Behaviors**						
MVPA	3 (605)	3.55 [3.29, 3.81]	***p* < 0.00001**	I^2^ = 97% (*p* < 0.00001)	2.546 (*p* = 0.238)	IcPAB Group
Step Count	2 (516)	0.15 [−0.19, 0.50]	*p* = 0.39	I^2^ = 57% (*p* = 0.13)	N/A	IcPAB Group
SB ^4^	2 (498)	1.10 [−1.19, 3.39]	*p* = 0.35	I^2^ = 99% (*p* < 0.00001);	N/A	IcPAB Group
**Health Outcomes**						
Test Anxiety	2 (68)	0.16 [−0.31, 0.64]	*p* = 0.50	I^2^ = 0% (*p* = 0.71)	N/A	IcPAB Group

^1^ Number of included study samples, ^2^ Sample size, ^3^ Standard Mean Difference, ^4^ Sedentary Behavior, ^APS^ Result after performing a sensitivity analysis, N/A: Not Applicable.

## Data Availability

Additional data supporting reported results can be found via Appendix A.

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
