# Peer review of "Effects of In-Classroom Physical Activity Breaks on Children’s Academic Performance, Cognition, Health Behaviours and Health Outcomes: A Systematic Review and Meta-Analysis of Randomised Controlled Trials"

_ijerph, 2022, doi:10.3390/ijerph19159479_

Round 1
Reviewer 1 Report
I congratulate the authors for the good systematic review and meta-analysis. The PROSPERO and COVIDENCE software were used and nine articles were investigated thoroughly. Random effects models were used to calculate the effect sizes. Few studies used theoretical frameworks and process evaluations. Health behaviors such as moderate-to-vigorous physical activity levels were improved, but IcPAB did not have an impact on cognition outcomes and health outcomes. I think the authors should indicate the ethnicities of individuals examined in both systematic review and meta-analysis. In the discussion, the contribution of diet and vitamin D supplementation should be discussed, being the reviewer a vitamin D researcher.
Author Response
Point 1: I congratulate the authors for the good systematic review and meta-analysis. The PROSPERO and COVIDENCE software were used and nine articles were investigated thoroughly. Random effects models were used to calculate the effect sizes. Few studies used theoretical frameworks and process evaluations. Health behaviors such as moderate-to-vigorous physical activity levels were improved, but IcPAB did not have an impact on cognition outcomes and health outcomes. I think the authors should indicate the ethnicities of individuals examined in both systematic review and meta-analysis. In the discussion, the contribution of diet and vitamin D supplementation should be discussed, being the reviewer a vitamin D researcher.
Response 1: Thank you very much for your comments and for appreciating our efforts in following the guidelines provided by the Cochrane Handbook for conducting systematic reviews and meta-analyses.
Regarding the ethnicities of individuals: Thank you very much for raising this point. Actually, only one study mentioned that 100% of the sample consisted of African American children (Layne, 2021). Other studies have not mentioned detailed information about the ethnicities of the sample. We have added this information to the text. Please see lines 227-229, line 406.
Regarding the diet and vitamin D supplementation: Thank you very much for directing our focus on this aspect as well. Actually, when setting PICO criteria for this systematic review we only decided to focus on the physical activity break interventions conducted within the classroom setting. And, none of the selected articles analysed the effects of IcPAB on students’ diet or Vitamin D supplements or how the diet/ vitamin D would affect the IcPAB or on any other outcome. Therefore, we have added this point to the discussion part. Please refer to lines 426 to 429.
Reviewer 2 Report
This is a review paper on the effects of breaks in classroom teaching in order for the children to do some kind of physical activity (IcPAB). The review is restricted to randomized control trials. The systematic database search identifies ten eligible studies. The authors claim that their meta-analysis find statistically significant effects on spelling performance, foreign language learning and some health-related behaviors.
While meta-analyses are useful to get a better understanding of the literature, it is important to acknowledge that such analyses also must make a critical judgement of degrees of freedom. Ten independent observations are below what is recommended, and the present case is even worse because the studies use different outcomes. It is illustrative that the estimated effects do not seem to be correlated with the intensity of the intervention. Thus, in my view, this paper is not publishable.
The paper could take a different turn by providing a qualitative discussion of the literature. Such an approach would naturally include a critical discussion about features of the interventions themselves. In this way it could provide some substance in recommendations for further research, both in terms of design of interventions and means to help improving the understanding of mechanisms driving potential positive effects.
Author Response
Point 1: While meta-analyses are useful to get a better understanding of the literature, it is important to acknowledge that such analyses also must make a critical judgement of degrees of freedom. Ten independent observations are below what is recommended, and the present case is even worse because the studies use different outcomes. It is illustrative that the estimated effects do not seem to be correlated with the intensity of the intervention. Thus, in my view, this paper is not publishable.
Response 1: We highly appreciate your critical comments on this work. Thank you very much for your thorough review and the comments.
Regarding the recommended observations: We have followed the guidelines and recommendations provided by the Cochrane Handbook. According to the guidelines, the minimum number of studies that are used to conduct a meta-analyse is two studies. The guidelines we referred to are mentioned in https://training.cochrane.org/handbook/current/chapter-10#section-10-2. In addition to that, it is justifiable that the total number of studies is 10, as there is a limited number of RCTs carried out to examine the effects of physical activity breaks, which are exclusive to the classroom. Furthermore, two published systematic reviews are good examples to justify the importance of our work, in which researchers also had less than 10 RCTs in their observations (Masini et al., 2020; Watson et al., 2017).
Masini A, Marini S, Gori D, Leoni E, Rochira A, Dallolio L. Evaluation of school-based interventions of active breaks in primary schools: A systematic review and meta-analysis. J Sci Med Sport [Internet]. 2020 Apr 1 [cited 2021 Mar 1];23(4):377–84. Available from: https://doi.org/10.1016/j.jsams.2019.10.008
Watson A, Timperio A, Brown H, Best K, Hesketh KD. Effect of classroom-based physical activity interventions on academic and physical activity outcomes: a systematic review and meta-analysis. Int J Behav Nutr Phys Act [Internet]. 2017 Dec 25 [cited 2021 Mar 1];14(1):114. Available from: /pmc/articles/PMC5574081/
Regarding the effects and intensity: Thank you for raising your concern on that. We agreed that intervention intensity is an important factor (moderator), which is correlated with the estimated intervention effects on each outcome variable. As the moderation examination is not our study purpose, we did not address this issue in our current manuscript. But we did seriously consider your constructive concern on that and made modifications accordingly from two aspects (1) elaborated new information about the total intensity of intervention both in Table 2 and in the text (see lines 239-241, 413-415). (2) recommended the future research direction regarding the moderation analyses of the total IcPAB intervention intensity (see lines 519-526).
Point 2: The paper could take a different turn by providing a qualitative discussion of the literature. Such an approach would naturally include a critical discussion about the features of the interventions themselves. In this way, it could provide some substance to recommendations for further research, both in terms of the design of interventions and means to help improve the understanding of mechanisms driving potential positive effects.
Response 2: Thank you very much for your comment. Based on your suggestion, we have amended the discussion part with more details. In particular, regarding the features of the interventions, we added information related to ethnicity, C-RCT and total intensity of intervention. Please see lines 406, lines 410-411, and lines 413-415. Regarding the design of intervention and means, we elaborated more information in the recommendations. We in-depth introduced the COM-B, a theoretical underpinning of IcPAB intervention, including its content and application, which will help improve the understanding of the mechanism resulting in potential positive effects. Please see lines 501-511.
Reviewer 3 Report
The authors conducted a very interesting and well-organized systematic review on the effects of in-classroom physical activy break on academic performance, cognition and healt outcomes in children. I have no comments regarding the manuscript except for line 360, where the sentence is missing. Please complete the phrase.
1.There is no comments on the compliance regarding the icPAB by the children and the teachers
2. Moreover, there was no comments regarding the effect of the icPAB on the loss of pure educational time
3. There was no comments whether icPAB may be applied in children with special educational needs.
Author Response
Point 1: The authors conducted a very interesting and well-organized systematic review on the effects of in-classroom physical activity breaks on academic performance, cognition and health outcomes in children. I have no comments regarding the manuscript except for line 360, where the sentence is missing. Please complete the phrase.
Response 1: Thank you very much for valuing our work, and pointing out the mistake in line 360. That line was a duplicate from the previous paragraph, and that might have happened when we were replacing our figures in the manuscript. We have deleted that line and thank you very much once again for pointing that out for us.
Point 2: There is no comments on the compliance regarding the icPAB by the children and the teachers.
Response 2: Thank you very much for pointing out this issue. When conducting the systematic review under the intervention’s fidelity we found that the researchers did not either mention teachers’ compliance or their strategies to enhance student engagement except for two studies (Berg et al., 2019; Egger et al., 2019). Berg’s and Egger's team mentioned the average number of IcPAB, which were implemented by the teachers. Based on your suggestions, we have added the compliance information in Table 2, the results part (lines 252-254), and the discussion part (lines 513-516).
Point 3: Moreover, there was no comments regarding the effect of the icPAB on the loss of pure educational time.
Response 3: Thank you for pointing out this issue. We agreed that the IcPAB might consume more of the pure educational time. However, many studies used IcPAB which was integrated into the curriculum. We have added this point to the discussion part. Please see lines 513-516.
Point 4: There was no comments whether icPAB may be applied in children with special educational needs.
Response 4: Thank you very much for raising this. In our PICO criteria, we defined that we were not focusing on the students with special education needs. This interesting point could be addressed in future research. We have added this to the discussion accordingly. Please see lines 488-489, 525-526.
Round 2
Reviewer 2 Report
The revised paper includes more information about the individual studies included in the meta-analysis. This is useful for the interpretation of the results. Since the number of studies included are limited, it is useful with more qualitative information, as in a traditional review paper.
My main concern with the paper remains. The fact that only ten experimental studies exist on the topic cannot be overcome. This is the reason why I initially suggested to reject the paper.
In the response letter, the authors argue, referring to a handbook, that “According to the guidelines, the minimum number of studies that are used to conduct a meta-analyse is two studies”, and further that “it is justifiable that the total number of studies is 10”. Technically, it is possible to calculate as many statistics as the number of observations. If one is interested in the mean and the standard error, as in the present paper, it is technically possible to estimate these two statistics based on only two observations. But this is clearly not recommended within applied social science. The handbook the authors refer to does neither recommend this, but state that “Meta-regression should generally not be considered when there are fewer than ten studies in a meta-analysis.” The handbook does not argue why ten is the magic number. In general, the confidence of findings increases in the number of observations. Using ten as a rule of thumb is clearly below what is used elsewhere in applied social science.
In the present paper, no one of the statistics is based on all ten studies. The statistics are based on 2-6 observations. This is because the studies use different outcomes. As far as I can understand, the meta-analysis in the present paper does not follow the recommendation of the handbook the authors refer to in their response letter. This can be illustrated by an example. The paper concludes that there are positive effects on spelling skills. This result, however, is based on one single study with “two different intervention arms”. The two arms are treated as two observations, which implies that the statistical conclusion is based on one study and two observations. In my view, it is obvious that we cannot not draw scientific meta-conclusions on such limited observation.
Author Response
Point 1:
The revised paper includes more information about the individual studies included in the meta-analysis. This is useful for the interpretation of the results. Since the number of studies included are limited, it is useful with more qualitative information, as in a traditional review paper.
[Response] Thank you very much for your critical review last time. Due to that, our paper was improved a lot as we could include more qualitative elaborations.
Point 2:
My main concern with the paper remains. The fact that only ten experimental studies exist on the topic cannot be overcome. This is the reason why I initially suggested to reject the paper.
In the response letter, the authors argue, referring to a handbook, that “According to the guidelines, the minimum number of studies that are used to conduct a meta-analysis is two studies”, and further that “it is justifiable that the total number of studies is 10”. Technically, it is possible to calculate as many statistics as the number of observations. If one is interested in the mean and the standard error, as in the present paper, it is technically possible to estimate these two statistics based on only two observations. But this is clearly not recommended within applied social science. The handbook the authors refer to does neither recommend this, but state that “Meta-regression should generally not be considered when there are fewer than ten studies in a meta-analysis.” The handbook does not argue why ten is the magic number. In general, the confidence of findings increases in the number of observations. Using ten as a rule of thumb is clearly below what is used elsewhere in applied social science.
[Response] Thank you very much for your insights. We agree with you, that it is possible to estimate pooled effect sizes based on two studies. And this is often not an impediment to pooling studies, the limitations just need to be acknowledged. That is why we mentioned that we did not perform meta-regression as there were limited number of studies. And we have acknowledged the limitation of this study, which was the fewer number of studies.
Actually, in addition to Cochrane Handbook, based on the literature (Pigott, 2012; Valentine et al., 2010) we learnt that there is no rule of thumb regarding the number of sufficient studies for a proper meta-analysis in applied social sciences. There is a long discussion on that among Research Gate scholars as well (https://www.researchgate.net/post/Is_there_a_minimum_number_of_articles_required_to_assess_publication_bias_while_conducting_a_meta-analysis).
Thus, we learnt that: 1) we should only be careful to avoid introducing selection bias into our literature search strategy; 2) Every study that meets selection criteria have to be analyzed; and 3) more studies equals to shorter confidence intervals and higher statistical power, as usual; Therefore 4) We must acknowledge the limitations.
Pigott’s book "Advances in Meta-Analysis" mentioned (Pigott 2012): “Another common question is: How many studies do I need to conduct a meta-analysis? Though my colleagues and I have often answered “two” (Valentine et al., 2010), the more complete answer lies in understanding the power of the statistical tests in meta-analysis. I take the approach in this book that power of tests in meta-analysis like power of any statistical test needs to be computed a priori, using assumptions about the size of an important effect in a given context, and the typical sample sizes used in a given field. Again, deep substantive knowledge of a research literature is critical for a reviewer in order to make reasonable assumptions about parameters needed for power (Pigott, 2012).
Furthermore, we have used random effects model to mitigate the limitation of having less than five studies in our meta-analysis. “If found less than five studies analyzing one topic or studies that were substantially heterogeneous, a random-effects model can be used in accordance with the Cochrane Handbook for Systematic Reviews of Interventions following the method of DerSimonian and Laird to compute the random-effects estimates for the corresponding statistics” (DerSimonian & Laird, 1986; Higgins et al., 2003).
Moreover, based on the literature review, we found that there have been numerous meta-analysis studies with less than 10 observations were published in high-impact peer-reviewed journals in applied social sciences discipline. This supports that it is an acceptable practice to conduct meta-analysis with less than ten observations in the applied social sciences domain. The eight examples of these meta-analysis papers are presented as follow for your reference.
- Masini, A., Marini, S., Gori, D., Leoni, E., Rochira, A., & Dallolio, L. (2020). Evaluation of school-based interventions of active breaks in primary schools: A systematic review and meta-analysis. Journal of Science and Medicine in Sport, 23(4), 377–384. https://doi.org/10.1016/j.jsams.2019.10.008 (JCR IF = 4.597, 17/87 in Sport Sciences).
(meta-analyzed the effects of some health behaviors by using 2 studies)
- Pyman, P., Collins, S. E., Muggli, E., Testa, R., & Anderson, P. J. (2021). Cognitive and Behavioural Attention in Children with Low-Moderate and Heavy Doses of Prenatal Alcohol Exposure: a Systematic Review and Meta-analysis. Neuropsychology Review, 31(4), 610–627. https://doi.org/10.1007/s11065-021-09490-8 (JCR IF =4.894, 2/67 in Neuropsychology and Physiological Psychology).
(meta-analyzed the effects of children’s behavioral attention by using 3 to 4 studies under each outcome)
- Lee, E. K., Donley, G., Ciesielski, T. H., Gill, I., Yamoah, O., Roche, A., Martinez, R., & Freedman, D. A. (2022). Health outcomes in redlined versus non-redlined neighborhoods: A systematic review and meta-analysis. Social Science & Medicine, 294, 114696. https://doi.org/10.1016/j.socscimed.2021.114696 (JCR IF = 5.379, 4/ 46 in Social Sciences, Biomedical).
(meta-analyzed the effects of health outcomes of women living in redlined areas by using 3 studies)
- Marker, C., Gnambs, T., & Appel, M. (2022). Exploring the myth of the chubby gamer: A meta-analysis on sedentary video gaming and body mass. Social Science & Medicine, 301, 112325. https://doi.org/10.1016/J.SOCSCIMED.2019.05.030 (JCR IF = 5.379, 4/ 46 in Social Sciences, Biomedical).
(meta-analyzed the effects of video gaming on physical activity by using 4 studies).
- Smith, S. R., Kroon, J., Schwarzer, R., & Hamilton, K. (2020). Parental social-cognitive correlates of preschoolers’ oral hygiene behavior: A systematic review and meta-analysis. Social Science & Medicine, 264, 113322. https://doi.org/10.1016/J.SOCSCIMED.2020.113322 (JCR IF = 5.379, 4/ 46 in Social Sciences, Biomedical).
(meta-analyzed the effects of the parents’ sense of coherence on pre-schoolers’ oral hygine by using 5 studies)
- Dworschak, C., Heim, E., & Maercker, A. (2022). Efficacy of internet-based interventions for common mental disorder symptoms and psychosocial problems in older adults: A systematic review and meta-analysis. Internet Interventions, 27, 100498. https://doi.org/10.1016/j.invent.2022.100498 (JCR IF = 5.358,17/ 109 in Health Care Sciences and Services).
(meta-analyzed the effects of the internet-based interventions on anxiety symptom severity in older adults by using 3 studies)
- Bolinski, F., Boumparis, N., Kleiboer, A., Cuijpers, P., Ebert, D. D., & Riper, H. (2020). The effect of e-mental health interventions on academic performance in university and college students: A meta-analysis of randomized controlled trials. Internet Interventions, 20, 100321. https://doi.org/10.1016/J.INVENT.2020.100321 (JCR IF = 5.358,17/ 109 in Health Care Sciences and Services).
(meta-analyzed the effects of e-mental health interventions on alcohol consumption by using 2 studies)
- Polak, M., Tanzer, N. K., Bauernhofer, K., & Andersson, G. (2021). Disorder-specific internet-based cognitive-behavioral therapy in treating panic disorder, comorbid symptoms and improving quality of life: A meta-analytic evaluation of randomized controlled trials. Internet Interventions, 24, 100364. https://doi.org/10.1016/J.INVENT.2021.100364 (JCR IF = 5.358,17/ 109 in Health Care Sciences and Services).
(meta-analyzed the effects of Disorder-specific internet-based cognitive-behavioral therapy by using 3 studies)
Furthermore, Turner and colleagues (Turner et al., 2013) said: “Most meta-analyses include data from one or more small studies… at least two adequately powered studies are available in meta-analyses reported by Cochrane reviews, underpowered studies often contribute little information, … However, underpowered studies made up the entirety of the evidence in most Cochrane reviews.” Turner and colleagues, make it clear that the lion’s potions of Cochrane reviews are meta-analyzed with fewer number of observations. Those studies may demonstrate an underpower. However, evidence from those studies cannot be disregarded as those also adding knowledge to the academia.
References
DerSimonian, R., & Laird, N. (1986). Meta-analysis in clinical trials. Controlled Clinical Trials, 7(3), 177–188. https://doi.org/10.1016/0197-2456(86)90046-2
Higgins, J. P. T., Thompson, S. G., Deeks, J. J., & Altman, D. G. (2003). Measuring inconsistency in meta-analyses. In British Medical Journal (Vol. 327, Issue 7414, pp. 557–560). BMJ Publishing Group. https://doi.org/10.1136/bmj.327.7414.557
Pigott, T. D. (2012). Advances in meta-analysis. In Advances in Meta-Analysis. Springer US. https://doi.org/10.1007/978-1-4614-2278-5/COVER
Turner, R. M., Bird, S. M., & Higgins, J. P. T. (2013). The Impact of Study Size on Meta-analyses: Examination of Underpowered Studies in Cochrane Reviews. PLoS ONE, 8(3), e59202. https://doi.org/10.1371/journal.pone.0059202
Valentine, J. C., Pigott, T. D., & Rothstein, H. R. (2010). How Many Studies Do You Need? Journal of Educational and Behavioral Statistics, 35(2), 215–247. https://doi.org/10.3102/1076998609346961
Point 3:
In the present paper, no one of the statistics is based on all ten studies. The statistics are based on 2-6 observations. This is because the studies use different outcomes. As far as I can understand, the meta-analysis in the present paper does not follow the recommendation of the handbook the authors refer to in their response letter. This can be illustrated by an example. The paper concludes that there are positive effects on spelling skills. This result, however, is based on one single study with “two different intervention arms”. The two arms are treated as two observations, which implies that the statistical conclusion is based on one study and two observations. In my view, it is obvious that we cannot not draw scientific meta-conclusions on such limited observation.
[Response] Thank you very much for your comment. We still would like to suggest that it is fair enough to conduct the meta-analysis with limited studies as we have clearly mentioned to the readers that the results of the study must be interpreted with caution due to limited number of studies. Hence, as you pointed out we do not directly suggest the readers come to a generalized meta-conclusion based on the limited number of studies. And that is why we also recommended the need for more RCT studies in this filed in future to obtain scientific meta-conclusions with more observations.
Furthermore, we would like to submit the same response that we have provided under point 2. to justify this matter. According to the handbook we have referred to, it is possible to conduct a meta-analysis by threating two arms of a single study as two different studies. Section 16.5.4 of the handbook says: “A further possibility is to include each pair-wise comparison separately, but with shared intervention groups divided out approximately evenly among the comparisons. For example, if a trial compares 121 patients receiving acupuncture with 124 patients receiving sham acupuncture and 117 patients receiving no acupuncture, then two comparisons (of, say, 61 ‘acupuncture’ against 124 ‘sham acupuncture’, and of 60 ‘acupuncture’ against 117 ‘no intervention’) might be entered into the meta-analysis”. (https://handbook-5-1.cochrane.org/chapter_16/16_5_4_how_to_include_multiple_groups_from_one_study.htm)
In our analysis, we used this example to conduct our meta-analysis for the spelling performance. Also, Polak et al., (2021) used the treated a three-arm study as two separate observations in their meta-analysis and the total observations included were three. This study was published in Internet Interventions (JCR IF = 5.358,17/ 109 in Health Care Sciences and Services).
References:
Polak, M., Tanzer, N. K., Bauernhofer, K., & Andersson, G. (2021). Disorder-specific internet-based cognitive-behavioral therapy in treating panic disorder, comorbid symptoms and improving quality of life: A meta-analytic evaluation of randomized controlled trials. Internet Interventions, 24, 100364. https://doi.org/10.1016/J.INVENT.2021.100364
We are honored to receive your critical responses and submit our counterresponses. We are sincerely grateful for your extensive review and comments. This is an amazing learning process as there is no universal agreement on several methodological aspects in conducting meta-analysis as there are lots of discussions going on for and against both for the responses from your end and our end. Hence, we strongly believe that the meta-analysis is justifiable as the methodological limitations are clearly acknowledged in our review.